# The Influence of Personal Harmony Value on Temporal Order Perception

**DOI:** 10.3390/bs13060459

**Published:** 2023-06-01

**Authors:** Li Pan, Xiting Huang

**Affiliations:** 1Research Center for Psychology and Social Development, Southwest University, No. 2 Tiansheng Road, Beibei, Chongqing 400715, China; panlimdx@163.com; 2Faculty of Psychology, Southwest University, No. 2 Tiansheng Road, Beibei, Chongqing 400715, China

**Keywords:** values, temporal order judgment, harmony, self

## Abstract

Empirical studies have demonstrated that self-relevant information affects temporal order perception. Therefore, the question arises of whether personal values—which are the core components of the self—influence temporal order perception. To explore this problem, we chose harmony, one of the most common values in Chinese culture, as the starting point. First, the harmony scale was used to measure the harmony values of the participants, and the participants were divided into high- and low-harmony groups. The validity of the grouping was then verified using an implicit-association test. Furthermore, two temporal order judgment (TOJ) tasks were used to explore the impact of harmony values on temporal order perception. The results revealed that in both TOJ tasks, participants in the high-harmony group tended to perceive harmonious stimuli before non-harmonious stimuli, while the effect was not found in the low-harmony group. We conclude that harmony values affect temporal order perception, and only if the values are important to the individual.

## 1. Introduction

Personal values are broadly desirable goals that motivate people’s actions and serve as guiding principles in their lives. Values affect people’s perception, cognition, and behavior over time and across situations [1]. So, do personal values affect temporal order perception? Temporal order perception is a type of time perception, which refers to the perception of the simultaneity, succession, and sequence of events within tens or hundreds of milliseconds. The temporal order judgment (TOJ) task is the most commonly used research paradigm in temporal order perception, and is widely used in both single modality and cross modality studies [2,3,4]. In the visual TOJ task, two visual stimuli are usually presented, one after another or simultaneously, within a short time frame, and participants are asked to determine which stimulus appears first [5,6]. The point of subjective simultaneity (PSS) is one of the common indicators used in TOJ task. PSS is an estimate of the time interval within which one sensory event had to lead another event in order for synchrony to be perceived [7].

Values are closely linked to the self. Verplanken (2002) [8] affirmed that a value affects behavioral decisions only when it is central to the self-concept. Studies based on implicit-association tests revealed that participants responded faster and more appropriately when words with a high importance-level value were paired with self-attribute words, while low-importance-level values tended to be incompatible with self-attribute words at the implicit level [9]. Neuroscience research has also revealed that important values often have a high degree of representation consistency with the self-concept. For example, in the study by Brosch et al. (2012) [10], participants were asked to perform two tasks: thinking about how important a value was for them as guiding principle in their life (core value condition), or how much they liked performing the activity (economic value condition). The results of brain imaging revealed that the medial prefrontal cortex (MPFC) was activated under the core value condition, compared with the economic value condition. MPFC is a brain region closely associated with self-related information processing [11]. It can be seen that personal values are closely related to the self.

Gallagher (2000) [12] collated evidence from empirical studies in multiple domains to classify the self as “narrative self” and “minimal self”, based on the presence or absence of temporal malleability. The minimal self is a primitive, immediate, and direct subject of experience, excluding all non-essential features of the self, and does not have a temporal extension, including the sense of agency (SoA) and the sense of ownership (SO).

A few studies based on the SoA found that in the TOJ task, participants reported that the stimuli caused by them appeared earlier than the stimuli caused by others. For example, in the study of Haering and Kiesel (2012) [13], red and yellow stimuli appeared immediately or 50 ms after the participant pressed a key. When participants were told that one participant caused one of the two stimuli, while the other participant seated at the other computer caused the other stimulus, they tended to perceive that the stimuli caused by themselves appeared first. Studies based on the SO also found that in a TOJ task, stimuli belonging to the participants themselves are more likely to be perceived earlier than the stimuli belonging to others. For example, in the study of Truong et al. (2017) [14], participants learned whether everyday objects belonged to them (self-owned) or the experimenter (other-owned), and completed a TOJ task. Results revealed a prior-entry effect, wherein participants were more likely to report that self-owned objects occurred earlier than other-owned objects. This effect was also found in the study by Constable et al. (2018) [15]. In their study, they found a reliable change in participants’ PSS in favor of their own objects. These results revealed that self-relevant information affects temporal-order perception. Based on the previous review, it is clear that personal values are closely related to the self, and the more important the values, the closer the connection to the self. The self affects temporal-order perception, and self-related stimuli are more likely to be perceived as appearing first, compared to non-self-related stimuli. Therefore, we speculate that personal values may affect temporal order perception. Specifically, in a TOJ task, high-importance stimuli are more likely to be perceived to occur earlier than low-importance stimuli.

Chinese people hold a deep-rooted desire to pursue “harmony” when cultivating one’s self, handling interpersonal matters, and confronting the universe and nature. In China, many proverbs show Chinese people’s emphasis on harmony values. For example, “家和万事兴” (harmony in a family makes everything successful), “和气生财” (harmony generates wealth), and so on. A few empirical studies have demonstrated that harmony values rank highly in the Chinese personal values system [16,17]. The Chinese focus on harmony also shows a tendency to avoid conflict. One study found that Chinese people reported higher levels of conflict avoidance compared with Americans [18]. Leung et al. (2011) [19] used a scale to measure harmony motivation among Chinese and Australian people, and found that Chinese people scored higher on harmony motivation, indicating that the concept of harmony is more important in Chinese societies than in Western societies. In summary, “harmony” is one of the core values of Chinese culture. Therefore, this study investigated whether there was a stable PSS bias for harmony-related stimuli in a TOJ task in Chinese. We first measured the level of harmony value of the participants through the harmony scale and implicit-association test (IAT) procedures. Next, we used two TOJ tasks to ascertain whether harmony values have an impact on temporal order perception. Based on the previous review, our research hypothesis was that participants would perceive harmonious stimuli as occurring earlier than non-harmonious stimuli if they regard harmony as an important value.

The stimuli presented to the participants in the TOJ tasks were usually simple graphics. In Chinese, the writing of harmony- and disharmony-related words is quite complicated, so we believed that it might not be appropriate to use Chinese characters as stimuli for order perception. The perceptual matching task can help to solve this problem. In the perceptual matching task used in a previous study, participants were asked to associate an abstract graph with a certain label, and then use the graph as a stimulus for subsequent research [20]. Therefore, a perceptual matching task was used in this study to associate harmonious and non-harmonious stimuli with relevant graphs, and on this basis, the influence of individuals’ harmonious values on temporal-order perception was examined.

## 2. Materials and Methods

### 2.1. Participants

This study was approved by the Ethics Committee of Southwest University. A group of 53 undergraduates were initially recruited to participate in the experiment. Four participants performed unsatisfactorily in the TOJ tasks and were excluded, and 49 valid participants were retained (33 were women; age range was 18–27 years, *M* = 21.80 years). All participants were right-handed. Participants had normal or corrected-to-normal vision.

### 2.2. Stimuli and Apparatus

Materials for measuring explicit harmony values. Harmony values were measured by the harmony enhancement subscale of the Harmony Scale [19]. This scale consists of 13 items, which measure the degree to which subjects regard harmony as an important goal. Participants were asked to rate the items on a 5-point scale, ranging from 1 (strongly disagree) to 5 (strongly agree). The higher the score, the more importance the participants were believed to attach to harmony values. In this study Cronbach’s α was 0.77.

Materials for measuring implicit harmony values. The IAT procedure was used to measure implicit harmony values. The material contained target words and attribute words. The words were all from a previous open survey. According to the results of the open survey, the words with the highest frequency were selected as the materials of the IAT procedure. The target words included the six Chinese words related to harmony, and six words related to non-harmony. Words related to harmony were: “和谐” (harmony), “和睦” (concord), “融洽” (in tune), “协调” (coordinate), “友好” (friendly), and “祥和” (auspicious and harmonious). Words related to non-harmony were: “冲突” (conflict), “矛盾” (contradiction), “对立” (oppose), “摩擦” (friction), “争执” (dispute), and “对抗” (confrontation). The attribute words included six agreement-related words, and six disagreement-related words. The agreement-related words were: “赞同” (agree), “同意” (consent), “赞成” (approve), “满意” (satisfaction), “接受” (accept), and “支持” (support), while the disagreement-related words included “反对” (oppose), “拒绝” (reject), “排斥” (repulsion), “不悦” (dissatisfaction), “否认” (deny), and “抗拒” (resist).

Materials for TOJ tasks. In the shape-matching TOJ task, the targets were a black cir-cle and a black triangle of 1.5° in width and 1.5° high, presented randomly at two different positions, at a 4.8° visual angle from the fixation on a white background. The fixation point consisted of a centrally presented black fixation cross (0.5° × 0.5° visual angle). In the color-matching TOJ task, the targets were a red circle and a green circle of 1.5° in width and 1.5° high.

### 2.3. Design and Procedure

The experiment was divided into two parts. Part A involved the measurement of harmony values, and included explicit questionnaire measurement and IAT procedure measurement. Half of the participants performed explicit measurements first, and implicit measurements later, while the other half did the opposite. Part B included TOJ tasks, including the shape-matching TOJ task and the color-matching TOJ task. Half of the participants completed the shape-matching TOJ first, and then the color-matching TOJ, while the other half did the opposite. Some participants completed Part A first and then Part B, while others did the opposite. The E-Prime 1.0 software was used for programming and timing the experimental operations.

Procedure of implicit harmony value measures. The participants were instructed to classify a series of stimulus words into the target category, and two attributive categories. The IAT procedure consisted of seven stages. In the first stage, the F key was pressed when the agreement-related word appeared, and the J key was pressed when the disagreement-related word appeared (12 trials). In the second stage, the participants pressed the F key when they saw the harmonious stimulus, but pressed the J key when they saw the non-harmonious stimulus (12 trials). In the third stage, the F key was pressed when the harmony words or agreement-related words appeared, while the J key was pressed when the non-harmony words or disagreement-related words appeared (24 trials). The first three stages were practice stages, and the fourth stage was the formal test of the third stage (96 trials). The fifth stage was the practice stage, in which the participants were asked to press the F key when the non-harmony words appeared, and press the J key when the harmony words appeared. The sixth stage was also the practice stage. At this stage, the F key was pressed when the non-harmony words or agreement-related words appeared, while the J key was pressed when the harmony words or disagreement-related words appeared (24 trials). The seventh stage was the formal test of the sixth stage (96 trials). Each incorrect response was signaled by a red X centered below the stimulus word for 150 ms, while a correct response was signaled by a green O centered below the stimulus word for 150 ms.

Procedure of TOJs. Before the TOJ task, the participants were asked to complete the perceptual-matching task. Taking the shape-matching task as an example, the participants were first told that the circle represented the harmonious stimulus, while the triangle represented the non-harmonious stimulus, and then the participants carried out the shape-matching task. In each trial, a fixation point of 500–1000 ms was first presented on the screen. Subsequently, a pairing of a shape and label (harmony or non-harmony) was presented for 2000 ms. Two geometric shapes (triangle and circle, each 1.5° × 1.5°) were presented above the fixation at the center of the screen. The word of harmony or non-harmony (3.1°/3.6° × 1.6°) was displayed below the fixation. The angular distance between the center of the shape or the word, and the fixation cross, was 3.5°. All stimuli were shown on a white background. Participants judged whether the pairings of shape and label matched. Feedback (correct or incorrect) was given on the screen for 500 ms at the end of each trial. Each participant performed 120 trials. The color-matching task was the same procedure, except that in the color-matching task, the green circle represented the harmonious stimulus, while the red circle represented the non-harmonious stimulus.

In each trial of the TOJ tasks, a fixation point was presented at the center of the screen for 500–1000 ms, and then a stimulus appeared on one side of the fixation point (left or right). After a variable interval, the second stimulus appeared on the other side of the fixation point. Once the second stimulus had appeared, the image remained rendered for 1000 ms, before disappearing. A question mark then appeared at the center of the screen, asking the participants to judge which stimulus appeared first. In the shape-matching TOJ, participants were asked to judge which shape stimulus appeared first, while in the color-matching TOJ, they were asked to judge which color stimulus appeared first. Stimulus-onset asynchrony (SOA) had nine levels at ±120 ms, ±90 ms, ±60 ms, ±30 ms, and 0 ms. The SOA was positive for the first circle (or green circle), and negative for the first triangle (or red circle). The SOA was 0 when the stimulus appeared simultaneously on both sides. Half of the participants attempted the shape-matching TOJ, followed by color-matching TOJ. Meanwhile, the other half of the participants attempted the TOJ in reverse order. The response key was balanced between the different participants. Half of the participants pressed “D” when the circle (or green circle) side stimulus appeared earlier, and pressed “K” when the triangle (or red circle) side stimulus appeared earlier. Each condition was repeated 15 times. The total number of trials was 270. Participants had some practice before the official experiment. The TOJ task process is depicted in Figure 1.

## 3. Results

The harmony values questionnaire was analyzed, and the one-sample *t*-test was conducted for the score (*M* = 3.72, *SD* = 0.39), and the median value 3. The results revealed a significant difference between the score and the median value (*t*(48) = 13.00, *p* < 0.001, *d* = 1.86), indicating that participants regarded harmony as an important value. Each participant’s level of harmonious values was then assessed by comparing the difference between the harmony scores of the participants and the median value 3. It was found that 33 participants had scores significantly higher than 3, and these participants were classified in the high-harmony group (*M* = 3.89, *SD* = 0.30). The other 14 participants whose scores were not significantly different from 3 were placed in the low-harmony group (*M* = 3.30, *SD* = 0.24). An independent samples *t*-test was conducted on the harmony values scores of the two groups, and the results revealed a significant difference in the scores of the two groups (*t*(47) = 6.72, *p* < 0.001, *d* = 2.13). This indicated that participants in the high-harmony group hold harmony values in higher esteem than those in the low-harmony group do.

In accordance with the recommendations of Greenwald et al. (1998) [21], first, participants whose response time was lower than 300 ms were recorded as 300 ms, and those whose response time was higher than 3000 ms were recorded as 3000 ms. In addition, participants whose error rate was higher than 20% were eliminated. According to this criterion, no participants were excluded. The mean response time and standard deviation of each participant in the compatible- and incompatible-task phases were calculated. On this basis, the d-prime of the implicit effect was calculated, and a one-sample *t*-test was performed on this value (*M* = 1.38, *SD* = 0.32) of all participants, and the results showed significant difference from 0 (*t*(48) = 30.60, *p* < 0.001, *d* = 4.37), which indicated that participants had implicit harmony values. A one-sample *t*-test was conducted on the implicit effects of the high-harmony group (*M* = 1.48, *SD* = 0.24) and the low-harmony group (*M* = 1.14, *SD* = 0.37). The results demonstrated that both the high-harmony group (*t*(34) = 37.20, *p* < 0.001, *d* = 0.28) and the low-harmony group (*t*(13) = 11.60, *p* < 0.001, *d* = 3.11) were significantly different from 0. An independent samples *t*-test was conducted on the d-prime of the implicit effect of the two groups. The results showed that the implicit effect size of the participants in the high-harmony group was significantly larger than that of the low-harmony group (*t*(47) = 3.88, *p* < 0.001, *d* = 1.23), which indicated that the classification of high- and low-harmony groups was effective.

The participants’ accuracy reached 0.95 in both perceptual-matching tasks, which indicated that the participants indeed combined the shape with the value words. An in-dependent samples *t*-test was conducted on the accuracy of participants in the high- and low-harmony groups. The results revealed that there was no significant difference between the performance of the two groups in either the shape-matching task (*t*(47) = −1.45, *p* = 0.16, *d* = −0.46) or the color-matching task (*t*(47) = −0.48, *p* = 0.63, *d* = −0.15).

For the shape-matching TOJ task-data analysis, first, E-prime was used to calculate the correct responses under each SOA. Then, MATLAB software was used to calculate the point of subjective simultaneity (PSS). The psychometric function was obtained by calculating the percentages of “harmony first” (circle first) responses as a function of SOA (Figure 2), and a cumulative Gaussian function was fitted to the data to account for individual variability [22,23,24]. PSS was measured, and it corresponded to the amount of time by which one stimulus had to precede or follow the other to be perceived as occurring simultaneously. It was taken as the 50% point from the TOJ curve. In every condition, psychometric functions were calculated individually for each participant. If the fit of the function was poor, and the *R*^2^ value was less than 0.75 [25], the data were discarded. Using these criteria, four participants were excluded from the analysis, and only the data from 49 participants were valid and analyzed.

We conducted a one-sample *t*-test on the participants’ PSS against a test value of 0 to determine the PSS shift between stimuli [15,26]. The results showed that the PSS of all the participants (*M* = −3.13, *SD* = 7.77) were significantly different from 0 ms (*t*(48) = −2.82, *p* < 0.01, *d* = −0.40, *BF*_10_ = 5.19 (moderate evidence for H_1_)). We then explored the relationship between participants’ harmony scores and the PSS; the correlation between the D score and PSS was significant (*r* (49) = −0.41, *p* < 0.01), and the correlation between the Harmony Scale score and PSS was marginally significant (*r* (49) = −0.27, *p* = 0.06). The PSS of participants in the high-harmony and low-harmony groups were then analyzed separately. The results showed that the PSS of the high-harmony group (*M* = −5.09, *SD* = 7.78) was significantly different from 0 ms (*t*(34) = −3.87, *p* < 0.001, *d* = −0.66, *BF*_10_ = 63.00 (very strong evidence for H_1_)), while the PSS of the low-harmony group (*M* = 1.77, *SD* = 5.33) was not significantly different from 0 ms (*t*(13) = 1.24, *p* = 0.24, *d* = 0.33, *BF*_10_ = 0.52 (anecdotal evidence for H_0_)).

The color-matching TOJ task-data analysis method was consistent with the shape-matching TOJ task. The psychometric function was obtained by calculating the percentages of “green first” responses as a function of SOA (Figure 3), and the accumulative Gaussian function was used to fit the data. The PSS was taken as the maximum of the best-fitting TOJ curve.

A single-sample *t*-test was performed on the PSS. The results showed that the PSS of all the participants (*M* = −3.55, *SD* = 8.99) were significantly different from 0 ms (*t*(48) = −2.77, *p* < 0.01, *d* = −0.40, *BF*_10_ = 4.59 (moderate evidence for H_1_)). We then explored the relationship between participants’ harmony scores and the PSS; the correlation between d-prime and PSS was significant (*r* (49) = −0.51, *p* < 0.001), and the correlation between the Harmony Scale score and the PSS was significant (*r* (49) = −0.32, *p* < 0.05). The PSS of participants in the high-harmony and low-harmony groups were then analyzed separately. The results revealed that the PSS of the participants with high-harmony values (*M* = −5.72, *SD* = 8.94) was significantly different from 0 ms (*t*(34) = −3.79, *p* <0.001, *d* = −0.64, *BF*_10_ = 50.56 (very strong evidence for H_1_)), while the PSS of the participants with low-harmony values (*M* = 1.87, *SD* = 6.71) had no significant difference from 0 ms (*t*(13) = 1.04, *p* = 0.32, *d* = 0.28, *BF*_10_ = 0.43 (anecdotal evidence for H_0_)).

We conducted a mixed-factor ANOVA with the perceptual-matching type (shape-matching or color-matching) as a within-subject factor and the harmony group (high-harmony group or low-harmony group) as a between-subject factor. The results revealed that the main effect of the matching type was not significant (*F*(1, 47) = 0.03, *p* = 0.86, $\eta_{p}^{2}$ = 0.001). The main effect of the harmony group was significant (*F*(1, 47) = 13.42, *p* < 0.001,$\eta_{p}^{2}$ = 0.22). The PSS of participants in the high-harmony group was significantly lower than that of the low-harmony group. The interaction between the two factors was not significant (*F*(1, 47) = 0.06, *p* = 0.81,$\eta_{p}^{2}$ = 0.001).

## 4. Discussion

The present experiments were designed to evaluate the effect of an individual’s values on their temporal-order perception. We first used the harmony questionnaire to distinguish two groups of participants; one group of participants had high harmony values, while the other group of participants placed relatively low importance on harmony values. An IAT procedure was used to verify the rationality of the grouping. We then used two TOJ tasks to explore the effect of harmony values on temporal-order perception. The results revealed that during the two TOJ tasks, the participants in the high-harmony group tended to perceive the harmonious stimulus as occurring earlier than the non-harmonious stimulus, while the participants in the low-harmony group did not exhibit this effect. This result confirms our hypothesis that personal values affect temporal perception, and that only when this type of value is an important value to the individual, will it have the prior-entry effect.

Several previous studies have confirmed the influence of the self on temporal-order perception. For example, West et al. (2009) [27] found that in the absence of spatial cues, fearful and angry faces were more likely to be perceived before neutral faces. Some researchers have found that in the TOJ task, compared with a friend’s face and an unfamiliar face, one’s own face was more likely to be perceived first [28]. In addition, participants tended to report that the stimuli caused by them appeared earlier than the stimuli caused by others [13]. Even the stimuli belonging to themselves were more likely to be perceived earlier than the stimuli belonging to others [14,15]. We found that values affect temporal order perception. Values are the core of the self. Therefore, we essentially verified the results of previous studies; that is, the self will affect temporal order perception. Why does the self affect temporal order perception? Previous studies have shown that affective salience stimuli influence selective attention [29,30]. Self-relevance is a definitional component of affective salience with processing advantages [8]. For example, people are sensitive to their name and face [31,32]. Humphreys and Sui (2015) [33] proposed a self-attention network, and argued that self-related information does have a differential impact on the allocation of attention, and can alter the saliency of a stimulus in a manner that mimics the effects of perceptual saliency. According to the perceptual latency model, it takes a certain perceptual latency for each stimulus-induced sensory signal to reach the central processor, and temporal order perception depends on the perceptual latency of the stimulus [34]. Owing to its processing advantage, self-related information has a shorter perceptual latency and thus, it arrives at the central processor earlier, and is perceived first. In this study, to the participants in the high-harmony group, harmony was an important value, and the harmonious stimuli were self-related information with affective salience characteristics, so these stimuli were more able to capture attention and were perceived to occur earlier than the non-harmonious stimuli. However, for participants in the low-harmony group, the harmonious stimulus was not an affective salience stimulus, so the harmony value did not affect their temporal order perception.

This study has a few shortcomings, which can be improved on in the follow-up research. For example, in the perceptual-matching task, we only associated green with harmonious information and red with disharmonious information. A more rigorous operation should consider the opposite condition. In addition, only harmony, a common value in Chinese culture, was selected in this study, and several more values should be selected in subsequent studies, to further verify the influence of values on temporal order perception.

## 5. Conclusions

This study used two TOJ tasks to explore the effect of harmony values on temporal order perception; the result revealed that values do affect temporal order perception, and only if the values are important to the individual.

## Figures and Tables

**Figure 1 behavsci-13-00459-f001:**
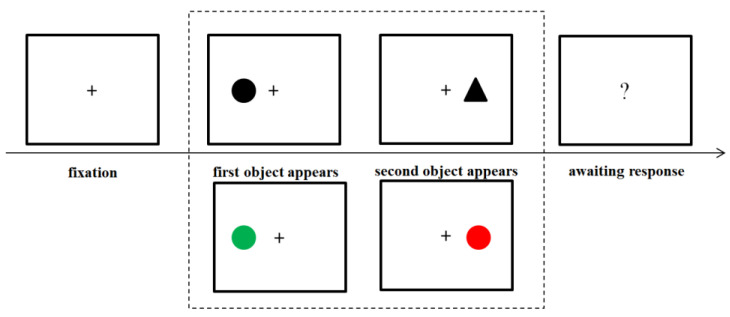
Typical trial sequences for the TOJ task, wherein participants were required to judge which target appeared first. In the shape-matching TOJ task, participants were asked to judge whether the circle or the triangle appeared first. In the color-matching TOJ task, participants were asked to judge whether the green circle or the red circle appeared earlier.

**Figure 2 behavsci-13-00459-f002:**
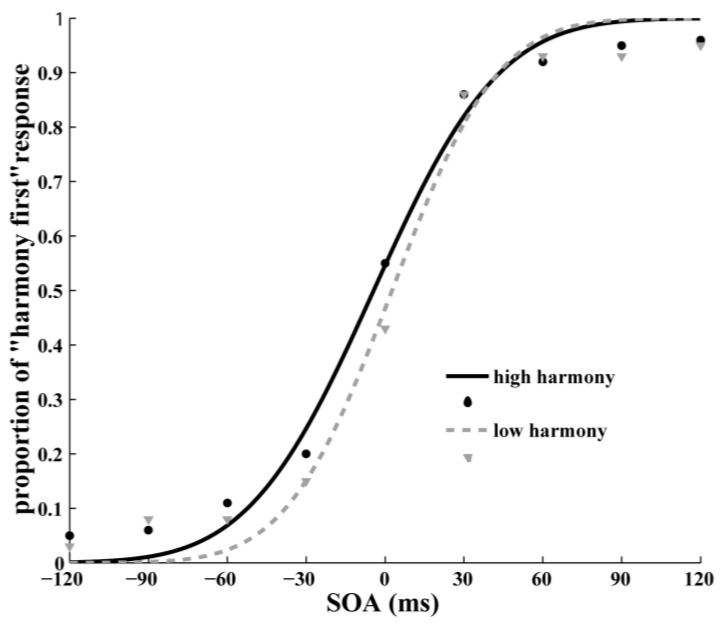
Data of shape-matching TOJ. The average proportion of “harmony first” response (circle first) at each SOA level was fitted with a cumulative Gaussian function.

**Figure 3 behavsci-13-00459-f003:**
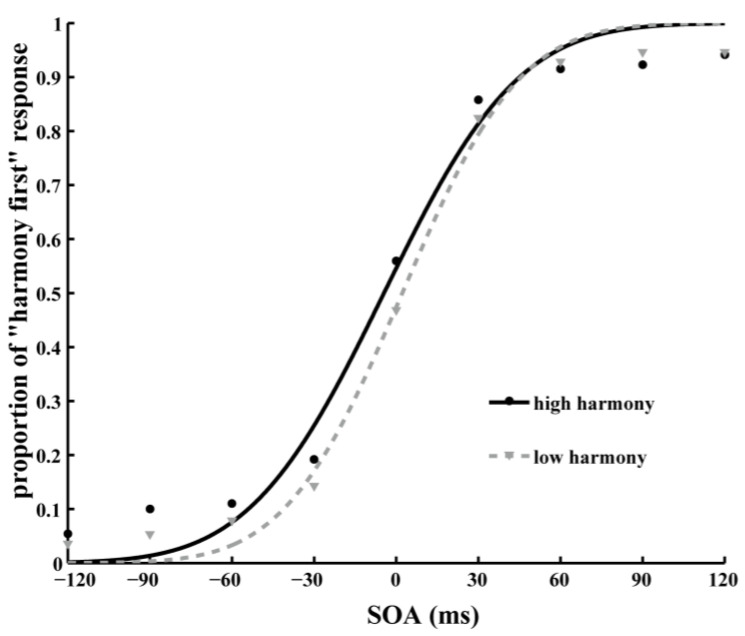
Data of color-matching TOJ. The average proportion of “harmony first” response (green circle first) at each SOA level was fitted with a cumulative Gaussian function.

## Data Availability

The data presented in this study are available on request from the corresponding author.

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
