# Peer review of "The Influence of Personal Harmony Value on Temporal Order Perception"

_behavsci, 2023, doi:10.3390/bs13060459_

Round 1

Reviewer 1 Report

Introduction

The introduction lacks an overview of TOJ. A paragraph explaining what TOJ is and its use should be added. Also, the authors should mention early in the introduction that they refer to TOJ in visual modality.

The connection between TOJ and personal values and the rationale to test their relationship should be strengthened.

Line 76: the influence of harmony values on temporal order perception. This is a correlational design that cannot point to causality. This statement should be rephrased to express the testing of an association rather than influence.

Line 75: “harmony” is the core value of Chinese culture. I don’t know the Chinese culture good enough to judge this statement, but I would feel safer if it was to be more modest. Maybe stating it is a core value rather than the one.

Method

Was verbal or written informed consent obtained? Both are mentioned. Written consent is good clinical practice.

Line 97: were initially recruited to participate. Wasn’t 29 the final sample size? If not, this should be mentioned here, along with reasons for drop-out.

The order of TOJ and harmony scale seems constant. This may have influenced the results by priming the participants to harmony.

Results

I don’t understand the aim of comparing the 4.14 score (was it the mean?) to 3 (on what basis it was decided to be median?). Later the median is said to be 4.14, which is very confusing.

I don’t understand the group comparisons for the TOJ tasks: from both figures, it doesn’t look like there are group differences. Also, the analysis is not clear. Why wasn’t there a group comparison instead of comparing each group to 0?

Author Response

  1. comment : The introduction lacks an overview of TOJ. A paragraph explaining what TOJ is and its use should be added. Also, the authors should mention early in the introduction that they refer to TOJ in visual modality.

response: Thank you very much for your valuable comments, and based on your suggestions, we have added the expression TOJ in the introduction section.

  1. comment :The connection between TOJ and personal values and the rationale to test their relationship should be strengthened.

response: We agreed with the suggestions of reviewer. In the definition of values, it was mentioned that values affect perception. We then wondered whether the particular perception of time perception would be affected by values. Unfortunately, no direct empirical studies were found to support the theory. Previous studies have shown that values are closely related to the self, and the more important the value, the more closely the value is related to the self. The self affects temporal perception, and specifically self-related stimuli are more likely to be perceived as occurring first compared to non-self-related stimuli. Therefore, we hypothesize that values affect temporal perception. Based on your suggestion, we have made some additions in the introduction section.

  1. comment : Line 76: the influence of harmony values on temporal order perception. This is a correlational design that cannot point to causality. This statement should be rephrased to express the testing of an association rather than influence.

response: We fully agree with your comments and have made revisions to the content.

  1. commentLine 75: “harmony” is the core value of Chinese culture. I don’t know the Chinese culture good enough to judge this statement, but I would feel safer if it was to be more modest. Maybe stating it is a core value rather than the one.

response: Thank you for pointing out the error here. Harmony is indeed one of the core values of Chinese culture. In the revised manuscript, we have revised it.

  1. commentWas verbal or written informed consent obtained? Both are mentioned. Written consent is good clinical practice.

response: Thanks for pointing out our mistakes. The participants provided their written informed consent to participate in this study.

  1. commentLine 97: were initially recruited to participate. Wasn’t 29 the final sample size? If not, this should be mentioned here, along with reasons for drop-out.

response: Thank the experts for your valuable suggestion. We added new participants to the revised manuscript, removed 4 invalid participants (R2 < 0.75 in the TOJ tasks), and finally retained 49 valid participants. The relevant expressions have been changed in the revised manuscript.

  1. commentThe order of TOJ and harmony scale seems constant. This may have influenced the results by priming the participants to harmony.

response: Yes, this suggestion is excellent, we did not consider this issue before, and in the revised version we added new participants. And this batch of participants completed the harmony measure after completing the TOJ task. In addition, we also tried to compare the PSS of the initial manuscript participants with those of the current supplemented participants, and there were no significant differences in the PSS for either the shape matching TOJ(t(47) = 0.90, p = 0.37) or the color matching TOJ tasks(t(47) = 0.80, p = 0.43). This indicates that the order of measurement did not affect the results.

Results

  1. commentI don’t understand the aim of comparing the 4.14 score (was it the mean?) to 3 (on what basis it was decided to be median?). Later the median is said to be 4.14, which is very confusing.

response: Thank you for your valuable comments. In the initial version of the manuscript, 4.14 was the mean value. The comparison with 3 was made because 3 was the median value considering that the scale is a 5-point scale. The median was 4.13 and it was considered that using the median was just enough to divide the participants into high and low harmony groups according to their scores. Now after your suggestion, we realized that using the median to divide high and low harmony groups is too crude.

In the revised version, on the one hand we analyzed all participants. On the other hand we compared each subject's scale score with the median value of 3. If it was significantly higher than 3, then the participant was classified into the high harmony group, and if it was not significantly different from the median (or significantly lower than the median), then it was classified into the low harmony group. Perhaps this classification is also not suitable enough.

  1. commentI don’t understand the group comparisons for the TOJ tasks: from both figures, it doesn’t look like there are group differences. Also, the analysis is not clear. Why wasn’t there a group comparison instead of comparing each group to 0?

response:Thank you again for your valuable comments. PSS is an estimate of the time interval by which one sensory event had to lead another event in order for synchrony to be perceived. For example, PSS=-3.55, means that when the non-harmonious stimulus appeared 3.55 ms before the harmonic stimulus, the participants perceived that the two stimuli appeared at the same time, while below 3.55 ms, the participants perceived that the harmonic stimulus appeared first, so PSS can reflect the perceptual tendency of the participants. Theoretically, the baseline level at which people perceive the simultaneous appearance of two stimuli is 0 ms, and therefore, the PSS is compared with 0. In addition, some researchers in temporal order perception studies have compared PSS with 0. The doi of two references are attached here.

doi: 10.1177/1747021818762010

doi: 10.1177/20416695211032993

    Based on your suggestion, we added the results of the group comparison in the revised manuscript. Also, because of the increased number of participants, the figure has been modified

Reviewer 2 Report

The current study investigated the effect of self-values on temporal order judgments. The authors target harmony as the self-value reference. Although the study is interesting, a few critical issues need to be addressed.

1. The argument for harmony in Chinese culture is not very relevant in the context of this paper, given the experimental design. After all, the participants are divided into those high and low-harmony groups, and no between-culture comparisons are made. The only data relevant to this culture issue is the mean scores on the harmony scale.   2. Statistical comparisons should be made between groups rather than separately comparing the groups to 0 ms. The entire data can be analyzed in one statistical model.   3. I do not understand why harmony was not treated as a continuous variance in the statistical analysis. The split-half method is not ideal; ideally, data should be treated as is.   4. The paper presents conflicting information regarding how the consent was given - is it verbal or written (check section 2.1 - second vs. last sentence)? Please indicate the protocol number.   5. It is not clear how many participants were excluded due to an error rate higher than 20%   "or-der" should be corrected as "order"

Author Response

1 comment. The argument for harmony in Chinese culture is not very relevant in the context of this paper, given the experimental design. After all, the participants are divided into those high and low-harmony groups, and no between-culture comparisons are made. The only data relevant to this culture issue is the mean scores on the harmony scale.

response:Thank you very much for your comments. The argument about harmony in Chinese culture in the introduction section was to explain that Chinese people have harmony values. In the revised version we analyzed the participants as a whole, and both the scores on the external harmony scale and the implicit D values indicated that Chinese participants regard harmony. Unfortunately we really did not make a comparison between cultures. Comparisons between cultures can be done in a follow-up study.

2 comment: Statistical comparisons should be made between groups rather than separately comparing the groups to 0 ms. The entire data can be analyzed in one statistical model.

response:This is indeed a very excellent suggestion!Based on your suggestion, we have added the corresponding content in the revised manuscript.

3 comment: I do not understand why harmony was not treated as a continuous variance in the statistical analysis. The split-half method is not ideal; ideally, data should be treated as is.

response:Thank you very much for your valuable comments. The unsatisfactory results of splitting into two in the first manuscript may be due to the insufficient amount of participants, and in the revised version, we added new participants and found the results of splitting into two to be ideal. In addition, based on your suggestion, we also considered harmony as a continuous variable, and in the revised manuscript we also analyzed all participants as a whole.

4 comment: The paper presents conflicting information regarding how the consent was given - is it verbal or written (check section 2.1 - second vs. last sentence)? Please indicate the protocol number.

response: Thanks for pointing out our mistakes. The participants provided their written informed consent to participate in this study.

 5 comment: It is not clear how many participants were excluded due to an error rate higher than 20%  "or-der" should be corrected as "order"

response: Thank you again for your recommendation that no participant was excluded in the IAT experiment for high error rate. Some errors have also been revised and corrected in the revised manuscript.

Reviewer 3 Report

This study used two temporal order judgment (TOJ) tasks to investigate the effect of personal harmony values on temporal order perception. Results indicated that participants with higher harmony values tended to perceive harmonious stimuli before non-harmonious stimuli, while this effect was not found in the low-harmony group. The authors conclude that harmony values affect temporal-order perception, and only if the values are important to the individual. 

The topic is interesting and the manuscript is overall well written. However, several concerns need to be addressed before further evaluation is considered.

Major points:

1. Dividing subjects into high and low harmony groups based on the median alone is too arbitrary, especially with this relatively small sample size. The authors should show the raw distributions of the explicit and implicit harmony values to see whether any of them are bimodally distributed. If not, the binary classification is inappropriate, even though the harmony values for these two groups are significantly different (lines 199 - 203), which is itself a result of double-dipping. A better approach may be to directly examine the correlation between harmony values and PSS in TOJ tasks.

2. Even with the binary classification, the effect size is too small to draw solid conclusions (5 ms and 3.6 ms PSS offset for the high harmony group in the shape and color task, respectively). For the color task, the p-value (0.054) is not even smaller than 0.05. At least a larger sample size is required.

3. It is unclear how the D value in the implicit harmony (line 209) was calculated. 

4. As the authors pointed out in the Discussion, the association between shape/color and harmony should vary across subjects/sessions. 

Minor points:

Line 30: should end with a comma
Line 106: what is coefficient alpha?
Line 139: two “agreement-related“s
Line 176: use the full name of the SOA on the first occurrence

Author Response

1 comment Dividing subjects into high and low harmony groups based on the median alone is too arbitrary, especially with this relatively small sample size. The authors should show the raw distributions of the explicit and implicit harmony values to see whether any of them are bimodally distributed. If not, the binary classification is inappropriate, even though the harmony values for these two groups are significantly different (lines 199 - 203), which is itself a result of double-dipping. A better approach may be to directly examine the correlation between harmony values and PSS in TOJ tasks.

response:Many thanks to the expert for your valuable comments. We checked the original distribution of harmony scores and indeed there is no bimodal distribution, and in the revised version we added the correlation analysis between PSS and harmony scores. It was found that the D-value of the IAT task reached a moderate negative correlation with both PSS, while the Harmony Scale score was weakly negatively correlated with the shape matching PSS and reached a moderate negative correlation with the color matching PSS.

2 comment: Even with the binary classification, the effect size is too small to draw solid conclusions (5 ms and 3.6 ms PSS offset for the high harmony group in the shape and color task, respectively). For the color task, the p-value (0.054) is not even smaller than 0.05. At least a larger sample size is required.

response:We strongly agree with your comments. In the revised manuscript, we have added new participants(Total valid participants were 49). After expanding the sample size, the high harmony group reached significant levels of PSS bias in both shape TOJ and color TOJ tasks. The Bayesian factor analysis provided very strong evidence in support of H1.

3 comment: It is unclear how the D value in the implicit harmony (line 209) was calculated.

response:Thank you for your comments. D values were calculated by referring to the study of Greenwald et al. (2003). We first calculated the mean response times of participant in the compatible and incompatible tasks, then obtained the difference in response times, and divided by the standard deviation of all trial response times in the two tasks. The doi of reference is attached here. DOI: 10.1037/0022-3514.85.2.197

4 comment: As the authors pointed out in the Discussion, the association between shape/color and harmony should vary across subjects/sessions.

response : We fully agree with your comment. This is indeed one of the shortcomings of this study.

  1. commentLine 30: should end with a comma

response: Thank you for pointing out our mistake, we have revised it.

  1. commentLine 106: what is coefficient alpha?

response: Thank you for your comment, what was intended to be expressed here was Cronbach's α, which has been revised.

  1. commentLine 139: two “agreement-related“s

response: Thank you for your comment, we have revised it  

  1. commentLine 176: use the full name of the SOA on the first occurrence

    response: Thank you for your comment, we have revised it

Round 2

Reviewer 1 Report

The revision is good. I have no further comments.

Author Response

Thank you so much for your previous work on our manuscript improvement.

Reviewer 2 Report

Authors have addressed my comments.

Author Response

(The authors gave the same response as above.)

Reviewer 3 Report

The revised manuscript is much improved and addresses most of my concerns. I have only two minor comments:

1. According to the authors' response, "D-value" is more commonly known as "d-prime". Please use "d-prime" instead.

2. For Cronbach's α, please specify "Cronbach's α", not just "α".

Author Response

Response to Reviewer 3

Comment1. According to the authors' response, "D-value" is more commonly known as "d-prime". Please use "d-prime" instead.

Response: Many thanks to the expert for your valuable comment. Based on your suggestions, we have revised the manuscript in the revised version.

Comment2. For Cronbach's α, please specify "Cronbach's α", not just "α".

Response: Thank you again for your valued comment. We strongly agree with your comment and have made changes in the revised manuscript.